# Adherence and Cost–Utility Analysis of Antiretroviral Treatment in People Living with HIV in a Specialized Clinic in Mexico City

**DOI:** 10.3390/pharmacy13030076

**Published:** 2025-05-28

**Authors:** Ivo Heyerdahl-Viau, Francisco Javier Prado-Galbarro, Santiago Ávila-Ríos, Osmar Adrian Rosas-Becerril, Raúl Adrián Cruz-Flores, Carlos Sánchez-Piedra, Juan Manuel Martínez-Núñez

**Affiliations:** 1Maestría en Ciencias Farmacéuticas, Universidad Autónoma Metropolitana Unidad Xochimilco, Mexico City 04960, Mexico; 2233800632@alumnos.xoc.uam.mx; 2Hospital Infantil de México “Federico Gómez”, Mexico City 06720, Mexico; frjavipg@gmail.com; 3Centro de Investigación en Enfermedades Infecciosas (CIENI), Instituto Nacional de Enfermedades Respiratorias (INER) Ismael Cosío Villegas, Mexico City 14080, Mexico; santiago.avila@cieni.org.mx; 4Department of Biological Systems, Universidad Autónoma Metropolitana Unidad Xochimilco, Mexico City 04960, Mexico; orosas@correo.xoc.uam.mx; 5Clínica Especializada Condesa Iztapalapa, Mexico City 09730, Mexico; racruz@sersalud.cdmx.gob.mx; 6Health Technology Assessment Agency (AETS), Instituto de Salud Carlos III (ISCIII), Madrid 28029, Spain; carlos.sanchez@isciii.es

**Keywords:** antiretroviral, HIV treatment adherence, HIV healthcare costs, CD4 cell count, viral suppression

## Abstract

This study aimed to evaluate the therapeutic adherence to antiretroviral therapy (ART) and the cost of care for people living with HIV (PLwHIV) in the Condesa Specialized Clinics (CSCs). A cross-sectional observational study was conducted using the Adherence Follow-Up Questionnaire developed by The AIDS Clinical Trials Group (ACTG) to measure adherence in 261 PLwHIV. An economic Markov model was developed to simulate clinical outcomes, health costs, and quality-adjusted life years (QALYs) over a 5-year horizon from the CSC perspective. The mean adherence index was 89.97, and 59% of the surveyed PLwHIV were non-adherent, but more than 95% of the population had an undetectable viral load, suggesting that ART remains effective in achieving clinical goals, even under suboptimal adherence conditions. More than half of the surveyed PLwHIV (60.54%) stated that they had stopped taking their ART at some point, and the three most frequent causes were forgetting (49.37%), being away from home (45.57%), and having a change in their daily routine (25.95%). The economic model showed a cumulative cost per PLwHIV of USD 8432 and 3.80 QALYs (USD 2218/QALYs), which is below the threshold of willingness to pay in Mexico (USD 13,790/QALY). These findings provide valuable information to guide public health decisions and resource allocation in HIV management in Mexico.

## 1. Introduction

Currently, infection with the human immunodeficiency virus (HIV) has become a manageable chronic condition, as people living with HIV (PLwHIV) can control their condition with antiretroviral therapy (ART). To date, ART is effective enough to provide a quality of life similar to that of the general population [1].

Since 2019, the medication Biktarvy^®^ (Gilead Sciences, Inc. Foster City, CA, USA) has been used as the preferred ART regimen in Mexico [2] and in other parts of the world [3]. This product combines the drugs bictegravir (BIC), emtricitabine (FTC) and tenofovir alafenamide (TAF) in a single tablet [2]. This medication has been shown to have greater effectiveness, fewer adverse effects, and lower risks of resistance compared to its predecessors [4]. However, for ART to be effective, there must be optimal therapeutic adherence [5], and, unfortunately, there are various problems that can hinder adherence to ART in the long term [1]. Non-adherence can lead to an increase in viral load, lower CD4 cell counts, and higher mortality from PLwHIV, as the infection worsens and diseases associated with acquired immunodeficiency syndrome (AIDS) develop [5]. For example, it has been estimated that PLwHIV in Latin American countries have an adherence to ART of 70%, meaning that 30% of the PLwHIV are at risk of suffering from disease progression [6].

Measuring adherence to ART represents many obstacles, and no official records are publicly available in Mexico. Additionally, according to the Antiretroviral Management Guide of People with HIV of Mexico, it is recommended to asses adherence at each medical visit, but this assessment is only mandatory in cases of virological failure [7]. Understanding the rates and determinants of non-adherence can help health systems improve pharmaceutical and medical care strategies, and can be the trigger for an active education and dissemination campaign that contributes to the promotion of therapeutic adherence.

On the other hand, the costs of medical care for this condition are varied and unpredictable; it is indisputable that its economic impact is relevant worldwide [8]. In this context, it is important to note that HIV prevalence in Mexico has increased as a result of fewer deaths among PLwHIV and increased incidence [9]. Additionally, almost half of PLwHIV have a late diagnosis [10], which leads to a greater consumption of health resources. Therefore, health systems must be prepared to face higher expenses. This represents a considerable challenge for public healthcare systems, which must allocate limited resources across multiple demands. Evaluating the economic impact of care for PLwHIV can provide valuable insights for the health systems to better understand the status of the economic burden of this disease and, with it, optimize healthcare spending and prepare for future expenditures.

The Condesa Specialized Clinics (CSCs) located in Mexico City provide public outpatient care and comprehensive medical care to PLwHIV. They serve over 20,000 PLwHIV [11], which represents ≈ 15% of Mexican PLwHIV on ART with an active status [12].

This study aimed to evaluate the adherence to ART among PLwHIV treated with the BIC/FTC/TAF scheme at one of the CSCs and to conduct a cost–utility analysis of this regimen, assessing the economic impact of medical care provided at the CSCs from an institutional perspective.

## 2. Materials and Methods

### 2.1. Study Design

This observational, cross-sectional, and descriptive study was conducted in a CSC during the last semester of 2024 to measure therapeutic adherence to ART in PLwHIV. Additionally, an economic evaluation was performed from the perspective of the CSCs to estimate the costs of HIV care.

### 2.2. Measuring Adherence to ART

#### 2.2.1. ART Adherence Follow-Up Questionnaire

The Spanish version of The Adherence Follow-Up Questionnaire developed by The AIDS Clinical Trials Group (ACTG) was used. This questionnaire is recommended in the Mexican Antiretroviral Management Guide for People with HIV to monitor therapeutic adherence. According to this guide, this questionnaire consists of a validated scale with the level of viremia and with satisfactory accuracy. It is available online for the medical personnel of the Ministry of Health in Mexico through the Antiretroviral Administration, Logistics and Surveillance System (SALVAR, in Spanish) [7].

This instrument evaluates whether PLwHIV take their doses as prescribed and evaluates compliance with special instructions such as monitoring the schedule, among other specifications. Additionally, it is possible to calculate the adherence index (AI) according to the equations and scores developed by Reynolds et al. (2007) [13]. Furthermore, this instrument identifies the reasons why PLwHIV may have stopped taking their medication.

#### 2.2.2. Selection of PLwHIV to Be Surveyed

PLwHIV who had been under the BIC/FTC/TAF treatment scheme for a minimum of one year at the time of application of the questionnaire were included. Cisgender and transgender men and women were included. Elimination criteria included incomplete or contradictory responses and unavailable clinical records.

The required sample size was 255 PLwHIV. The minimum sample size was calculated based on the classic equation for descriptive studies [14], based on an adherence proportion of 79% in Latin American countries (Mexico and Puerto Rico) [6], with a 95% confidence interval and a 5% margin of error. The study objectives and implications were explained to all participants, and they provided written informed consent before completing the questionnaire. The participants answered the questionnaire in a private setting, and the researchers were available to clarify any doubts.

#### 2.2.3. Consultation of Clinical Data of the Surveyed PLwHIV

Data on CD4 cell count, viral load, age, and time under treatment of the surveyed PLwHIV were retrieved from the SALVAR database at the CSC site.

#### 2.2.4. Data Categorization and Statistical Analysis of Results

The AI was calculated according to Reynolds et al. (2007) [13] for both PLwHIV and the overall average. According to the Mexican Antiretroviral Management Guide for People with HIV [7], optimal adherence should be equal to or greater than 95%; therefore, adherence was categorized as a dichotomous variable: adherent (≥95%) and non-adherent (<95%).

The time under treatment was categorized as 1 to 2 years, 3 to 4 years, and 5 years. The age of PLwHIV was categorized based on the median quartiles of our sample’s age distribution to divide the population into four subgroups of approximately equal size, ensuring balanced comparisons. Therefore, age was categorized as follows: <28 years, 28 to 34 years, 35 to 42 years, and >42 years. Viral load was categorized according to the World Health Organization (WHO) [15] and the Mexican Antiretroviral Management Guidelines for People with HIV [7] as undetectable (<50 copies/mL) and unsuppressed (>50 copies/mL). Finally, CD4 cell counts were categorized according to the WHO’s classification of immunological states of HIV infection as CD4 < 200 cells/μL, CD4 200–349 cells/μL, CD4 350–499 cells/μL, and CD4 ≥ 500 cells/μL [16].

#### 2.2.5. Logistic Regression

Logistic regression models were applied to assess factors associated with treatment adherence, considering the clinical and sociodemographic characteristics of PLwHIV. The outcome variable “adherence” was categorized into non-adherent and adherent. A *p*-value < 0.05 was considered statistically significant.

### 2.3. Cost of HIV Care

A non-comparative economic evaluation of the costs associated with the management of newly diagnosed HIV cases was conducted from the CSC perspective, using data from the surveyed PLwHIV.

#### 2.3.1. Economic Model

A Markov model (Figure 1) was constructed using TreeAge Pro^®^ software. The model included four clinical stages based on WHO classifications of CD4 levels [16]. Immunological status was selected as the primary stratification criteria since CD4 cell count correlates with quality of life in PLwHIV. The CD4 ≥ 500 cells/μL stage represents an asymptomatic phase with the highest quality of life, whereas lower CD4 levels indicate increased susceptibility to opportunistic infections and reduced well-being [17]. The stage of death was not considered in the model because the CSCs only provide outpatient care and because the model was based on real-world data from the surveyed PLwHIV.

At baseline, PLwHIV are distributed in the four different clinical stages depending on CD4 levels at ART initiation. Once ART is initiated, PLwHIV could transition between stages annually, either improving, remaining stable, or worsening in immunological status. Each cycle represented one year, and the time horizon of the model was five years.

#### 2.3.2. Model Inputs

The model was fed from the data obtained from the surveyed PLwHIV. The model entries are shown in Table 1 and are detailed in the following sections.

#### 2.3.3. Model Population

Because CD4 cell count progression differs between ART-experienced and ART-naïve PLwHIV [21], only the clinic data of the surveyed PLwHIV who initiated ART for the first time with BIC/FTC/TAF (n = 237) were included in the economic analysis. The baseline distribution of the model population by clinical stage was based on the epidemiology of incidence observed in the surveyed PLwHIV (Table 1).

#### 2.3.4. Costs

Costing was conducted from the CSCs perspective. A gross-costing approach was used to estimate medical care expenses. Only direct medical costs were considered. The health resources and services used in the care of PLwHIV were first identified based on the Antiretroviral Management Guide for People with HIV in Mexico [7] and the standard care provided at the CSCs. Considered resources included BIC/FTC/TAF ART, diagnostic and monitoring studies such as viral load, CD4 count, viral genotype and clinical laboratory tests, tests for sexually transmitted diseases, hepatitis B and C, and the GeneXpert MBT/RIF study. Other medical services such as medical consultations, imaging studies, and prophylactic antibiotics were also included.

Annual resource utilization per person among the PLwHIV was estimated for the first year of care and subsequent years, stratified by immunological stage. Data sources included the clinical reports from the CSCs, the Antiretroviral Management Guide for People with HIV in Mexico [7], and the expert opinions from healthcare providers at the clinic. The proportion of PLwHIV who suffered virological rebounds and consumed additional health resources more frequently, such as viral load studies, CD4 studies, and medical consultations, was also considered.

Subsequently, a monetary value was assigned to the identified resources. Unitary costs were obtained from the CSC records and public tenders in the Official Gazette of the Federation [19,20]. Costs were expressed in U.S. dollars (USD). The considered average exchange rate for 2024 was calculated with Banxico’s public data (USD 1 = MXN 18.22) [22].

The equation with which to calculate the total annual cost of HIV for each immunological stage consisted of adding the annual costs of each health resource used:$$Annual\;cost\;per\;PLwHIV\;per\;stage=\left({UC}_{1}\times F_{1}\times P_{1}\right)+\left({UC}_{2}\times F_{2}\times P_{2}\right)\ldots$$
where

UC = unitary cost of the health resource;

F = annual frequency of use of the health resource;

P = annual proportion of PLwHIV using the health resource.

This equation was used to determine the initial cost (of the first year of care) and the incremental costs (of subsequent years, individually). An annual discount rate of 5% for costs was considered, following the recommendations of the Guide for the Conduct of Economic Evaluations in Mexico [23]. The initial cost and incremental costs for each clinical stage are shown in Table 1. The Microsoft Excel file detailing the costing methodology is included in the Appendix A.

Finally, a willingness to pay (WTP) threshold of 1 gross domestic product per capita was established according to the Guide for the Conduct of Economic Evaluations in Mexico [23], and it is USD 13,790 [24].

#### 2.3.5. Utilities

Utility was represented by quality-adjusted life years (QALYs). The measure of QALYs depends on a utility score that represents people’s quality of life. This score ranges from 0 to 1, where 0 is death and 1 is perfect health [25]. Stage-specific utility scores were derived from Whitham et al.’s study (2020) [18] (Table 1). A 5% annual discount rate was applied to health benefits, as recommended by the Guide for the Conduct of Economic Evaluations in Mexico [23].

#### 2.3.6. Transition Probabilities

The model’s transition probabilities were obtained from the clinical data of the 237 surveyed PLwHIV who initiated ART with BIC/FTC/TAF. We considered the annual increases and decreases in CD4 levels throughout their clinical history to determine the proportions in which PLwHIV annually changed from one immunological stage to another. Due to insufficient data on PLwHIV at years 4 and 5, it was assumed that they had the same behavior as in the third year of care. The transition probabilities are shown in Table 2. The Microsoft Excel file detailing the methodology used to obtain the transition probabilities is provided in the Appendix A.

#### 2.3.7. Monte Carlo Simulation and Sensitivity Analysis of the Economic Model

To evaluate the robustness of the model, a probabilistic sensitivity analysis was performed with all the model inputs, i.e., values of unit costs, utilities, baseline probabilities, and transition probabilities. A variation of 7% was applied according to the Guide for the Conduct of Economic Evaluations in Mexico [23], except for the range of variation in utilities, which were obtained from the scientific literature [18,26].

A Monte Carlo simulation with 10,000 iterations was carried out to obtain the variability of cost and QALY outcomes. Additionally, a univariate sensitivity analysis was conducted on model variables to identify those with the greatest individual impact on the model and total healthcare costs.

### 2.4. Ethics

The study was carried out in accordance with the Declaration of Helsinki and was approved by the Ethics and Research Committee of the Metropolitan Autonomous University (ID number: CEI.2020.002).

## 3. Results

### 3.1. Study Sample

A total of 327 PLwHIV met the inclusion criteria, but only 270 agreed to participate in the study (response rate of 82.57%). Nine people were ruled out because their clinical reports were not available. The final sample was 261 PLwHIV (Figure 2).

Table 3 shows the general clinical characteristics of the PLwHIV included in the study. The majority were male (88.9%). Almost half had 1 to 2 years of experience with ART (49.8%). The vast majority did not have a history of ART switching (≈91%). It is noteworthy that the average AI was almost 90%, and 60% of the PLwHIV were non-adherents. Furthermore, the current mean CD4 levels were 502 cells/μL, and the most frequent immunological category was ≥500 cells/μL (44.44%), while the initial mean CD4 levels were 320.10 and almost 40% were in the worst immunological stage.

### 3.2. Therapeutic Adherence, Adherence Index and Causes of Non-Adherence

Table 4 shows the results of therapeutic adherence. There was no significant difference in therapeutic adherence between the evaluated groups.

Figure 3A shows that just over half of the PLwHIV (54.79%) always followed their medication schedule. Most PLwHIV (78.93%) stated that they were aware of the special instructions for taking their ART, of which 60.18% always followed them (Figure 3B). On the other hand, just over a third of the PLwHIV reported never having stopped taking their ART (39.46%) (Figure 3C), while the vast majority stated that they had not stopped taking their ART during the last weekend (93.8%) (Figure 3D).

### 3.3. Factors Associated with Treatment Adherence

Table 5 shows that PLwHIV with CD4 350–499 cells/μL are more likely to be adherent than those with CD4 < 200 cells/μL (OR = 3.32, 95% CI: 1.021–10.816). Additionally, PLwHIV who stopped taking their medication on the weekend were less likely to be adherent compared to those who did not stop taking it (OR = 0.17, 95% CI: 0.037–0.803).

More than half of the surveyed PLwHIV (60.54%) stated that they had stopped their ART at some point. Table 6 shows the reasons why the PLwHIV discontinued their treatment. The three most frequent causes were forgetting (49.37%), being away from home (45.57%), and having a change in their daily routine (25.95%).

### 3.4. Analysis of Clinical Progression in the Markov Cohort

Figure 4 shows the result of the simulation of disease progression in the hypothetical population, using the clinical characteristics of the surveyed PLwHIV. With regards to the progression of a hypothetical population through the immunological stages, it stands out that more than 70% of the population reaches a CD4 level ≥ 500 cells/μL in the fifth year, while the prevalence of CD4 stage < 200 cells/μL reaches almost zero at year 5 of treatment.

### 3.5. Costs of HIV Care

The model showed a five-year cumulative cost of USD 8431.95 per person among the PLwHIV and a cumulative effectiveness of 3.80 QALYs, i.e., USD 2218.93/QALY, which is below the threshold of WTP in Mexico (USD 13,790/QALY). Figure 5 shows the scatter plot derived from a probabilistic simulation of cost variability and QALYs. It highlights that all points are very close to the average for both costs and QALYs.

### 3.6. Sensitivity Analysis

The sensitivity analysis carried out to assess the robustness of the model is shown in Figure 6. It highlights the five variables that most significantly impact healthcare costs. A univariated analysis for each of these variables showed that, individually, the unitary cost of BIC/FTC/TAF affects the costs of care by ±5%, medical visits by ±0.78%, viral load studies by ±0.55%, CD4 studies by ±0.28%, and the proportion of PLwHIV initiating ART in CD4 < 200 cells/μL by ±0.2%.

## 4. Discussion

This study makes a significant contribution to clinical research related to PLwHIV, as it is, to our knowledge, the first study conducted in Mexico to examine both therapeutic adherence and healthcare costs from an institutional perspective. Additionally, this research is based on real-world clinical data from the target population, unlike most economic studies that rely on clinical trial data.

More than 250 PLwHIV completed the ACTG Follow-Up questionnaire. The vast majority of the surveyed PLwHIV were male, which aligns with the national epidemiology of HIV infection [27]. In addition, nearly 40% of PLwHIV started ART with an immune stage of CD4 < 200 cells/μL, which in Mexico is considered a late diagnosis [28]. This incidence is slightly lower compared to the latest epidemiology published at the national level (43%) [10]. Notably, similar figures have been reported over the past two decades. For example, in an epidemiological study of HIV infection in Mexico conducted between 2007 and 2014, a weighted incidence of late onset of 48.6% was estimated, according to SALVAR data [29]. In another very similar study covering 2008–2017, this incidence was estimated to be 44.93% [30]. Although care for HIV has improved in Mexico, there is an urgent need to implement strategies that promote early diagnosis.

The mean AI observed in this study was almost 90%, falling short of the optimal threshold. In fact, more than half of the surveyed PLwHIV were classified as non-adherent. Despite this, clinical data from 95% of people showed an undetectable viral load, which is slightly higher than the current national epidemiology (89% undetectability) [10]. Moreover, less than 10% have a CD4 immune stage of <200 cells/μL, and, in fact, the predominant CD4 category found was ≥500 cells/μL, which is associated with better quality of life and an asymptomatic status [31].

However, statistical analysis revealed no significant differences in AI across demographic and clinical variables, as all groups had a mean AI close to the overall average. This suggests that adherence is not strongly influenced by these factors and that self-reported adherence does not necessarily correlate with clinical outcomes. A possible explanation for this finding could be that the BIC/TAF/FTC regimen has a high resistance barrier due to the inclusion of second-generation integrase inhibitors, which contribute to both genetic robustness and rapid viral suppression. This provides the drug with strong forgiveness, meaning that even with suboptimal adherence, it remains effective [32]. This forgiveness is more favorable for this regimen than others. For example, the minimum requested adherence to maintain virological efficacy is 90% for the dual regimen Dolutegravir (DTG) + Lamivudina (3TC), while it is only 70% for BIC/FTC/TAF [33]. Although viral suppression is the ultimate clinical goal, and was indeed observed in our population, behavioral adherence remains essential for ensuring long-term treatment success. Therefore, adherence monitoring continues to be relevant even in populations with favorable clinical outcomes.

On the other hand, we found a significant association between adherence and having a current CD4 count of 350–499 cells/μL, but not ≥500 cells/μL. This may suggest a difference in behavior, as individuals within this intermediate range may still perceive themselves to be at clinical risk or under closer monitoring, which could motivate higher adherence. In contrast, those with CD4 ≥ 500 cells/μL may perceive themselves as healthy and stable, potentially leading to a relaxation in adherence behavior.

Another key finding relates to ART adherence patterns on weekends. Previous studies have reported that weekends often disrupt medication schedules, posing a risk to adherence [34], which is frequent when it comes to ART [35,36,37]. This contrasts with our results, as almost all of the surveyed PLwHIV did take their ART on the weekend. This may be because people tend to feel less busy and overwhelmed on the weekend, making forgetfulness less frequent. However, although it was not a frequent occurrence in the present study, logistic regression analysis showed that those who missed ART doses on weekends were significantly more likely to be classified as non-adherent. This is interesting, since Reynolds et al. (2007), who developed the equation to obtain the AI through the ACTG Follow-Up questionnaire, observed that this variable was redundant and was the only element of the survey that was not included in this equation [13].

A key strength of the ACTG Follow-Up questionnaire is that it assesses both medication intake and adherence to special instructions (e.g., taking ART at the same hour each day, and avoiding food–drug interactions), which are also considered part of therapeutic adherence. In this study, it is highlighted that most of the surveyed PLwHIV are aware of the special instructions, which suggests good communication with the physician. However, only 60% of PLwHIV who know the special instructions stated that they always follow them, which suggests that people may underestimate the importance of attending to them or have difficulties in their personal and daily lives. This is important because adhering to a certain schedule for taking ART is essential to maintain consistent plasma levels, avoiding resistance and virological failures, while taking care of co-administration with certain medications and foods allows PLwHIV to avoid interactions and absorption problems [38].

However, the algorithm with which to obtain the AI through this questionnaire gives a lot of weight to this variable, so, for example, if a person has never failed their ART, but does not follow the special instructions, their AI drops to 87%, far from the value that is considered optimal. Although it is important to follow the schedule instructions and avoid taking it with certain foods and medications, to our knowledge, there is no evidence that this variable affects biochemical outcomes so significantly, especially for a regimen such as BIC/FTC/TAF, which does not have as many special instructions as previous therapeutic regimens [39,40]. Consequently, the AI observed through this calculation may reflect the attitude and perception of PLwHIV towards their ART, but it will not necessarily correlate with their health status, and this may also be one of the reasons why there are no significant differences in the AI outcomes between groups.

This is combined with the fact that the primary reasons for ART discontinuation are preventable factors related to lifestyle (e.g., forgetting, being away from home, and changes in daily routine). Therefore, it is important to promote awareness of the relevance of adherence to ART among PLwHIV, for example through educational interventions, and to help the person adapt their treatment to their daily life in a personalized way, which has already given beneficial results for PLwHIV [41]. Psychosocial and emotional barriers such as stigmatization, depression, or even the feeling of being well had low frequency as causes of the lack of therapeutic adherence, which suggests good understanding and education about HIV infection among PLwHIV and in their environment. Additionally, very few respondents cited medication toxicity or side effects as reasons for non-adherence. This is consistent with previous findings that BIC/FTC/TAF has a low discontinuation rate due to adverse effects [42], which were much more frequent with previous antiretroviral regimens [43,44]. Overall, these data suggest that stopping the medication is usually unintentional and may be mainly due to lifestyles and not to the medication itself or the medical care received by PLwHIV.

Furthermore, the Markov cohort behavior suggests that ART and medical care at the CSCs are effective, as most PLwHIV progress to the highest immunological stage over time. However, 30% of the population does not reach CD4 ≥ 500 cells/μL within five years, indicating that some individuals experience slower immune recovery, a fact that can be deduced from the scarce change in the intermediate immunological stages of 200–349 cells/μL and 350–499 cells/μL.

In this regard, it is important to consider that there is a group of individuals known as immunological non-responders who do not recover normal CD4 levels even after prolonged ART. Previously reported risk factors that contribute to this scenario include being male, being of an older age, having co-infections, and having low CD4 cell counts [45]. Therefore, the stagnation at suboptimal immunological stages may have been because most PLwHIV were male and a significant proportion had late diagnoses, and not because of low adherence or ART inefficacy. This again emphasizes the need for early diagnosis. Therefore, healthcare providers should identify and prioritize interventions for these PLwHIV to help them achieve optimal CD4 restoration. However, it is important to note that more than 80% of PLwHIV reach CD4 levels of ≥350 cells/μL, which is already considered a clinically favorable immunological state for PLwHIV, associated with a significantly lower risk of opportunistic infections and improved long-term outcomes [45].

Regarding costs, in the Markov model, we observed a cost-effectiveness ratio below the WTP threshold for Mexico (USD 13,790). Therefore, if PLwHIV continue to respond in the same way to treatment, the CSCs can pay for their care. The sensitivity analysis and probabilistic simulation showed low variability in costs and effectiveness, with all iteration results located very close to the average values and below the WTP threshold. This confirms the robustness of the model and guarantees that the treatment will be affordable, even under conditions of uncertainty for all variables and worst-case scenarios. In this regard, it is important to notice that for CSCs, the monthly cost of BIC/FTC/TAF is only USD 94.40. This is because the Mexican public sector carries out consolidated purchases, negotiating directly with manufacturers and distributors, which allows the government to have preferential prices that are significantly lower than those of the retail market. For example, today, it is possible to find BIC/FTC/TAF at USD 752.82 in a Mexican private pharmacy [46].

Although the acquisition cost of BIC/FTC/TAF is low for the Mexican public sector, it is noteworthy that, in the sensitivity analysis, the unitary cost of this drug was the one that had the greatest impact on total healthcare costs, which was to be expected, since it has previously been observed that the incorporation of this regimen into health systems has represented an increase in the economic impact on the health sector [47,48]. Sensitivity analysis also highlights that the proportion of PLwHIV that begin in the worst immunological stage is the clinical and epidemiological characteristic that most affects the total costs of care, which again reinforces the need to promote earlier diagnosis.

Although healthcare is affordable, the costs can be substantial, and the public sector must be ready to afford the investment. For example, according to the latest public data, in the CSCs, it is estimated that 21,749 PLwHIV receive healthcare [49], so the total investment is around USD 36,677,274.11 per year. In addition, it is estimated that there is an annual growth rate of care of 12.1% [49]; therefore, healthcare expenditures are expected to rise significantly in the coming years, and could double to USD 64.75 million by 2029 if no structural changes are implemented. This raises questions about the long-term sustainability of financing, especially within a publicly funded healthcare system where resources are shared among competing priorities. Thus, early diagnosis and investment in prevention are essential to mitigate financial pressure on the public healthcare system.

This study has some limitations. First, it was a monocentric cross-sectional study, so it is advisable to carry out larger studies in other specialized care centers with longitudinal follow-ups. Similarly, these results are representative of Mexico City; however, they may not accurately reflect the situation in other parts of the country, where the epidemiology and management of the disease could differ. Also, because PLwHIV were approached when they went to the pharmacy for their medication, the sampling frame is likely biased toward PLwHIV with good adherence habits, which is reflected by the similar AI for all groups and, in most cases, the undetectable viral load. In addition, self-report questionnaires may be subject to bias. It is also important to consider that the odds ratios in factors associated with adherence presented wide confidence intervals, which may indicate limited statistical power for certain associations. Therefore, these results should be interpreted with caution, and future research with larger samples is recommended to validate these trends. In addition, the socioeconomic characteristics of PLwHIV, which could also influence therapeutic adherence, were not identified.

The great strength of the developed economic model lies in the use of real-world clinical data from the population served in the health center of interest, and not from clinical trials or secondary sources, providing realism and transparency to the obtained results. Therefore, the structure and methodology of the model (based on real-world immunological changes and real-world costs) can be adapted to different epidemiological profiles and other national or international settings where epidemiological data, medical costs, and health service usage patterns are available. Additionally, the results of the Markov cohort showed expected immunological recoveries, suggesting that the model represents realistic clinical outcomes. However, the transition probabilities between clinical stages were only obtained from the first three years of medical care for the surveyed PLwHIV. In addition, treatment costs for opportunistic infections were not considered, so costs may be underestimated. However, most PLwHIV had an undetectable viral load and were at an immunological stage greater than 350 cells/μL, suggesting a low risk of opportunistic infections. In addition, most commonly used antibiotics in CSCs have low cost, as they can be acquired by less than USD 1 per unit, which represents a negligible fraction of the total cost of care. In fact, CSCs are primary care centers, and when PLwHIV require more extensive treatments or hospitalization, they are referred to tertiary-level institutions such as the National Institute of Respiratory Diseases (INER in Spanish), which is the hospital that treats the largest number of PLwHIV in advanced stages in Mexico [50].

## 5. Conclusions

PLwHIV treated in the CSC have suboptimal adherence, with a mean adherence rate of 89.97%, and 59% of them were categorized as non-adherent. However, the clinical goals are achieved; more than 95% of the population maintain an undetectable viral load, and there is a clear tendency towards immune restoration, suggesting that the ACTG Follow-Up questionnaire mainly reflects people’s attitudes and perceptions towards their ART more than its clinical features. This also suggests that despite the suboptimal adherence levels, according to the questionnaire, BIC/FTC/TAF remains clinically effective. Economically, the cumulative cost per person among the PLwHIV over five years was USD$ 8431.95. For the current population of the CSCs, the estimated total annual cost is USD 36,677,274.11. The cost–utility ratio of USD 2218.93/QALY is below the threshold of WTP in Mexico, so the estimated costs for future care for this population are within the affordability range for Mexico’s public health sector. The unitary cost of BIC/FTC/TAF is the variable with the greatest impact on the total costs of care. However, it should be noted that the proportion of PLwHIV who initiate ART in a low immune state is also a variable that increases care costs and limits immune recovery, so the need for strategies for early diagnosis is emphasized. These findings provide key information for optimizing the use of resources in the public health system and highlight the importance of strengthening interventions to promote therapeutic adherence.

## Figures and Tables

**Figure 1 pharmacy-13-00076-f001:**
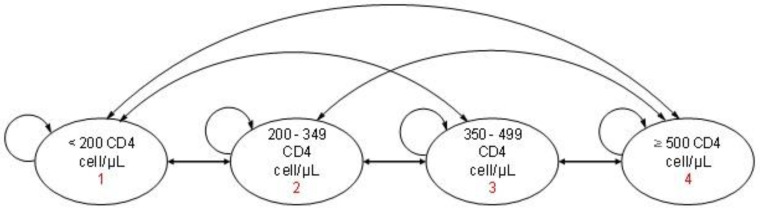
Markov model for the economic evaluation of ART in surveyed PLwHIV.

**Figure 2 pharmacy-13-00076-f002:**
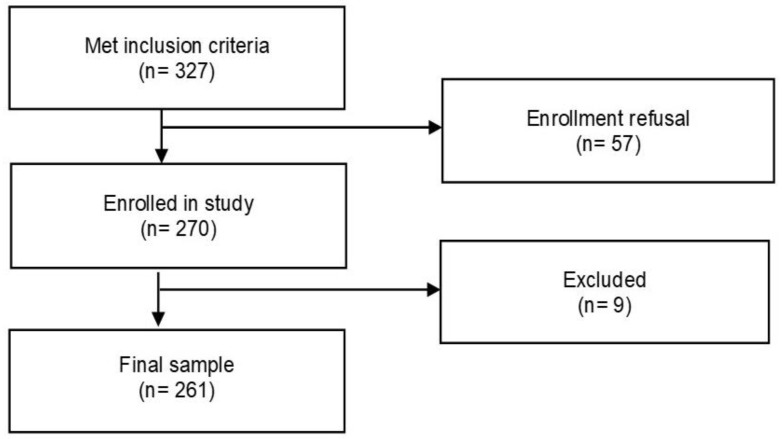
Flow chart of the study sampling process.

**Figure 3 pharmacy-13-00076-f003:**
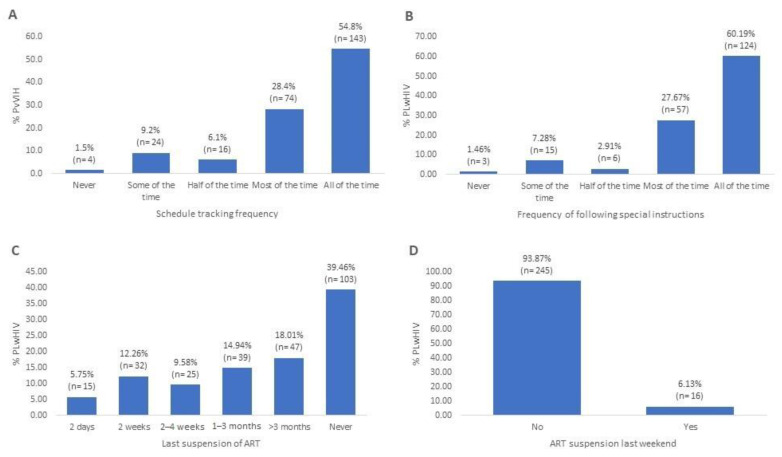
Adherence behavior in the studied population. (**A**) Proportion of PLwHIV who consistently followed their ART schedule (adherence to dosage timing). (**B**) Frequency with which PLwHIV who are aware of special ART instructions (e.g., taking medication at the same schedule or avoiding food interactions) report following them. (**C**) Time elapsed since the last reported interruption of ART by PLwHIV (current compliance behavior). (**D**) ART adherence during the most recent weekend (behavioral changes during non-working days).

**Figure 4 pharmacy-13-00076-f004:**
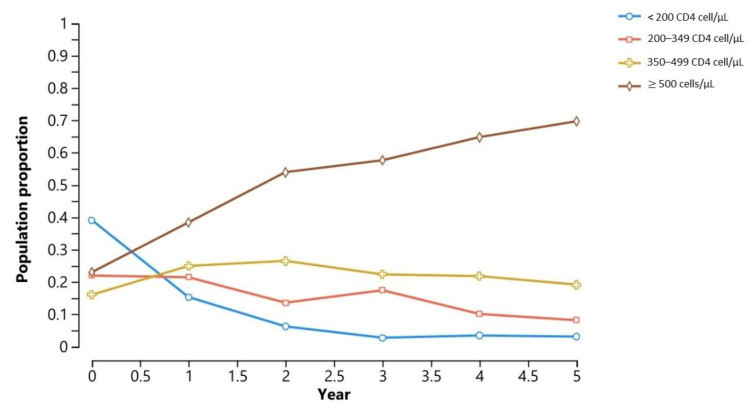
Temporal distribution of immunological stages according to the Markov model.

**Figure 5 pharmacy-13-00076-f005:**
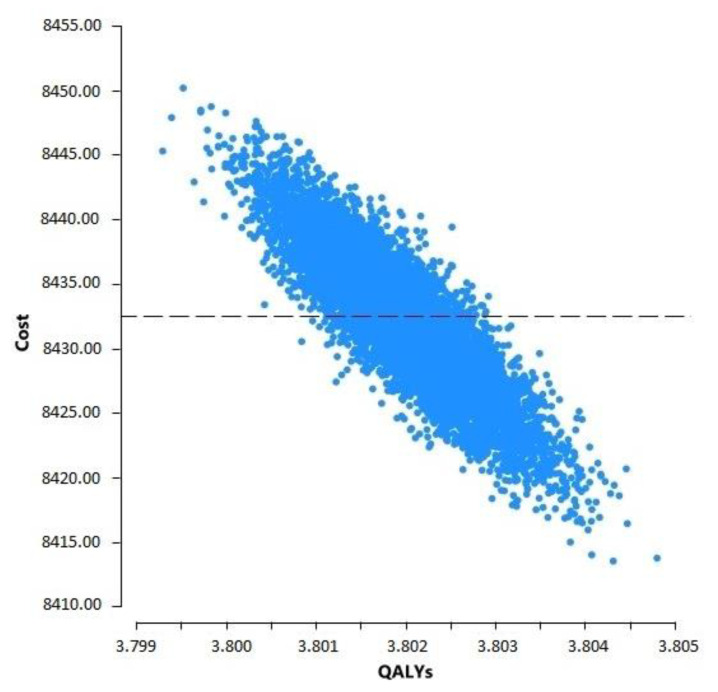
Scatter plot of cost and QALY variability of HIV care.

**Figure 6 pharmacy-13-00076-f006:**
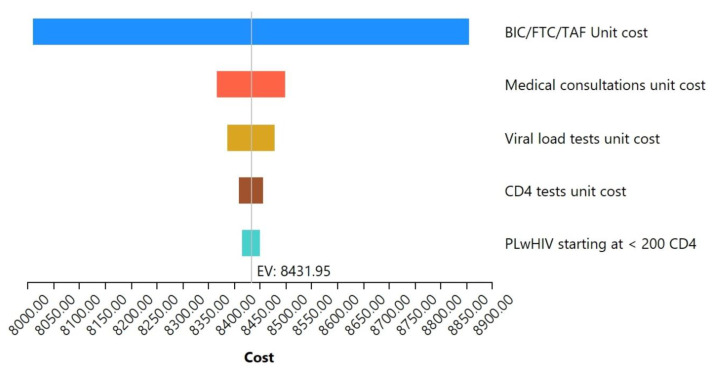
Multivariate sensitivity analysis. Tornado diagram of the five variables that most impact the total costs of HIV care.

**Table 1 pharmacy-13-00076-t001:** Markov economic model inputs.

Immune Stage	Baseline Probability	Utility Score (QALYs)	Initial Cost (Cycle 1) (USD$)	Incremental Cost (Cycles 2–5) (USD$)
CD4 < 200 cells/μL	0.39	0.67	2147.37	2025.54
CD4 200–349 cells/μL	0.22	0.70	2118.01	1440.75
CD4 350–499 cells/μL	0.16	0.71	2118.01	1406.49
CD4 ≥ 500 cells/μL	0.23	0.73	2118.01	1400.78
Sources	CSCs	Whitham, et al. (2020) [18]	CSCs and Official Gazette of the Federation [19,20]	CSCs and Official Gazette of the Federation [19,20]

**Table 2 pharmacy-13-00076-t002:** Markov economic model transition probabilities.

Transition Between CD4 Stages	Year 1	Year 2	Years 3–5
<200–<200	0.38	0.36	0.57
<200–200–349	0.36	0.61	0.43
<200–350–500	0.20	0.04	0.00
<200–500	0.07	0.00	0.00
<200–349–<200	0.02	0.03	0.11
<200–349–200–349	0.28	0.58	0.32
<200–349–350–500	0.30	0.28	0.43
<200–349–500	0.40	0.11	0.14
<350–500–<200	0.00	0.00	0.00
<350–500–200–349	0.05	0.18	0.10
<350–500–350–500	0.34	0.49	0.38
<350–500–500	0.61	0.33	0.52
<500–<200	0.00	0.00	0.00
<500–200–349	0.02	0.01	0.02
<500–350–500	0.22	0.10	0.10
<500–500	0.76	0.88	0.88

**Table 3 pharmacy-13-00076-t003:** Variables of the surveyed PLwHIV.

Variable	(N = 261) n (%)
Gender	
Male	232 (88.9%)
Female	20 (7.7%)
Transgender female	9 (3.4%)
Age (mean ± SD)	35.74 ± 9.43 years
<28 years	63 (24.1%)
28–34 years	63 (24.1%)
35–42 years	80 (30.7%)
>42 years	55 (21.1%)
History of ART switch	
With switch history	24 (9.2%)
No switch history	237 (90.8%)
Time with current ART (mean ± SD)	2.70 ± 1.28 years
1–2 years	130 (49.8%)
3–4 years	98 (37.6%)
5 years	33 (12.6%)
AI (mean ± SD)	89.97 ± 11.98
Non-adherents (AI < 95%)	154 (59.0%)
Adherents (AI ≥ 95%)	107 (41.0%)
Viral load	
Undetectable (<50 copies/mL)	249 (95.4%)
Unsuppressed (≥50 copies/mL)	12 (4.6%)
Current CD4 (mean ± SD)	502.11 ± 251.46
CD4 < 200 cel/µL	21 (8.1%)
CD4 200–349 cel/µL	56 (21.5%)
CD4 350–499 cel/µL	68 (26.0%)
CD4 ≥ 500 cel/µL	116 (44.4%)
Baseline CD4 (mean ± SD)	320.10 ± 238.01
CD4 < 200 cel/µL	102 (39.1%)
CD4 200–349 cel/µL	57 (21.8%)
CD4 350–499 cel/µL	47 (18.0%)
CD4 ≥ 500 cel/µL	55 (21.1%)

**Table 4 pharmacy-13-00076-t004:** Mean adherence index and adherence category by variable studied.

Variable	Mean AI ± SD	*p*-Value	<95% n (%)	≥95% n (%)	*p*-Value
Gender					
Male (n = 232)	89.92 ± 11.54		139 (59.9%)	93 (40.1%)	
Female (n = 20)	88.91 ± 17.92	0.616	11 (55.0%)	9 (45.0%)	0.606
Transgender female (n = 9)	93.61 ± 6.56		4 (44.4%)	5 (55.6%)	
Age					
<28 years (n = 63)	88.79 ± 12.79		38 (60.3%)	25 (39.7%)	
28–34 years (n = 63)	89.91 ± 12.43	0.623	40 (63.5%)	23 (36.5%)	0.778
35–42 years (n = 80)	90.16 ± 13.07		46 (57.5%)	34 (42.5%)	
>42 years (n = 55)	91.10 ± 8.57		30 (54.5%)	25 (45.4%)	
Time with current ART					
1–2 years (n = 130)	89.74 ± 10.94		83 (63.8%)	47 (36.1%)	
3–4 years (n = 98)	89.99 ± 14.24	0.497	51 (52.0%)	47 (48.0%)	0.196
5 years (n = 33)	90.80 ± 8.30		20 (60.6%)	13 (39.4%)	
Viral load					
Undetectable (<50 copies/mL) (n = 249)	89.90 ± 12.13		147 (59.0%)	102 (41.0%)	
Unsuppressed (≥50 copies/mL) (n = 12)	91.41 ± 8.76	0.744	7 (58.3%)	5 (41.7%)	0.961
Current CD4					
CD4 < 200 cells/μL (n = 21)	90.14 ± 5.64		16 (76.2%)	5 (23.8%)	
CD4 200–349 cells/μL (n = 56)	89.28 ± 14.49	0.741	34 (60.7%)	22 (39.3%)	0.348
CD4 350–499 cells/μL (n = 68)	90.28 ± 12.44		37 (54.4%)	31 (45.6%)	
CD4 ≥ 500 cells/μL (n = 116)	90.08 ± 11.33		67 (57.8%)	49 (42.2%)	
Initial CD4					
CD4 < 200 cells/μL (n = 102)	89.18 ± 12.54		65 (63.7%)	37 (36.3%)	
CD4 200–349 cells/μL (n = 57)	90.42 ± 13.32	0.215	29 (50.9%)	28 (49.1%)	0.198
CD4 350–499 cells/μL (n = 47)	92.86 ± 7.27		24 (51.1%)	23 (48.9%)	
CD4 ≥ 500 cells/μL (n = 55)	88.48 ± 12.55		36 (65.5%)	19 (34.6%)	

**Table 5 pharmacy-13-00076-t005:** Factors associated with ART adherence.

Variables	OR	95% CI	*p*-Value
Gender (Ref. Male)			
Female	1.190	0.454–3.120	0.724
Transgender female	1.576	0.396–6.276	0.518
Age (Ref. < 28 years)			
28–34 years	0.802	0.374–1.720	0.571
35–42 years	0.998	0.484–2.057	0.995
>42 years	1.279	0.576–2.838	0.545
Treatment time (Ref. 1 year)			
Treatment Time (2 years)	1.008	0.467–2.174	0.985
Treatment Time (3 years)	1.847	0.827–4.126	0.134
Treatment Time (4 years)	1.197	0.482–2.974	0.699
Treatment Time (5 years)	1.090	0.402–2.951	0.866
Viral load (Ref. Undetectable (<50 copies/mL))			
Unsuppressed (≥50 copies/mL)	1.012	0.293–3.500	0.985
Current CD4 (Ref. < 200 cells/μL)			
CD4 200–349 cells/μL	2.567	0.771–8.055	0.125
CD4 350–499 cells/μL	3.323	1.021–10.816	0.046 *
CD4 ≥ 500 cells/μL	2.882	0.918–9.046	0.070
Suspended ART over the weekend (Ref. No suspension)	0.173	0.037–0.803	0.025 *

* *p*-value < 0.05.

**Table 6 pharmacy-13-00076-t006:** Causes of ART discontinuation among surveyed PLwHIV.

Cause	n (%) (N = 158)
Simply forgot	78 (49.37%)
Was far from home	72 (45.57%)
Had a change in daily routine	41 (25.95%)
Busy with other things	39 (24.68%)
Ran out of pills	35 (22.15%)
Fell asleep through the dose time	32 (20.25%)
Had trouble taking the pills at a specific schedule	15 (9.49%)
Wanted to avoid side effects	10 (6.33%)
Felt good	10 (6.33%)
Did not want others to notice while taking medication	9 (5.70%)
Felt sick or unwell	8 (5.06%)
Had too many pills to take	7 (4.43%)
Felt depressed or overwhelmed	7 (4.43%)
Felt like the drug was toxic/harmful	2 (1.27%)

## Data Availability

The original contributions presented in this study are included in the article and the Appendix A. Further inquiries can be directed to the corresponding author.

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
