# Peer review of "Adherence and Cost–Utility Analysis of Antiretroviral Treatment in People Living with HIV in a Specialized Clinic in Mexico City"

_pharmacy, 2025, doi:10.3390/pharmacy13030076_

Round 1
Reviewer 1 Report
Comments and Suggestions for Authors
Overall, the manuscript is very relevant and presents clinical data that should be published. I offer just a few minor recommendations for improving the manuscript.
In Table 5, the confidence intervals are wide, suggesting low power in some comparisons. Is recommended to discuss this limitation.
It would be useful to include interactions between variables (e.g., initial CD4 and adherence).
The sustainability of long-term financing remains to be discussed, given the 12.1% annual growth in demand.
It would be useful to include a brief discussion on the transferability of the model to other contexts or regions.
Review orthotypographic punctuation (redundant use of spaces in some points).
Check consistency in the use of percentages and symbols (“%” vs. “percent”).
Author Response
We would like to express our sincere gratitude for the insightful review of our manuscript. The comments and suggestions provided have undoubtedly helped us improve the article's quality. All changes made to the manuscript in response to the observations have been indicated and are marked in yellow. We hope the new version meets your expectations.
Comment: In Table 5, the confidence intervals are wide, suggesting low power in some comparisons. Is recommended to discuss this limitation.
Response: Indeed, some of the confidence intervals in Table 5 are wide, which may reflect a limited statistical power. We have now acknowledged this issue in the limitations section, noting that some associations should be interpreted with caution.
Comment: It would be useful to include interactions between variables (e.g., initial CD4 and adherence).
Response: We have included “initial CD4” as a new variable in Table 4 “Mean adherence index and adherence category by variable studied” with its AI value and analysis.
Comment: The sustainability of long-term financing remains to be discussed, given the 12.1% annual growth in demand.
Response: It is a very good point, as it is certainly important to further discuss the sustainability of long-term financing given the 12.1% annual growth in demand. This point has been improved in the discussion.
Comment: It would be useful to include a brief discussion on the transferability of the model to other contexts or regions.
Response: We have discussed the transferability of the model to different contexts and regions in the strengths and limitations section of the discussion.
Comment: Review orthotypographic punctuation (redundant use of spaces in some points). Check consistency in the use of percentages and symbols (“%” vs. “percent”).
Response: We have reviewed orthotypographic punctuation and the use of symbols and percentages to make the text fully homogenous. Other language errors have also been reviewed.
Reviewer 2 Report
Comments and Suggestions for Authors
I am attaching my copy of the manuscript with some edits and comments. Some issues are:
- several tables have either "media" or "medium" when it should be "Mean" or "Median" (I can't tell from the data which is intended) - it needs clarification
- the breakdown of ages (<28, 28-34, 35-43, and >42) seems artificial. they're not even intervals in terms of years, and they don't yield even quartiles. choose one or the other.
- the statement that the AI goal is ≥95% is misleading; clinically the goal is viralogic suppression of ≥95% and you have that in your population, so the entire premise of the paper is not really appropriate. you should really be celebrating the fact that Biktarvy is performing so well (as it does, everywhere that it is used, due to ease of dosing, low toxicity profile, and high resistance barrier).
- Figure 3 - the legend simply restates the X-axes of each plot, rather than expanding and explaining. Not helpful in understanding.

most of the english is good, but there are a few errors which confuse the meaning, particularly in regards to the use of the words media and medium when the authors must mean either mean or median
Author Response
We would like to express our sincere gratitude for the insightful review of our manuscript. The comments and suggestions provided have undoubtedly helped us improve the article's quality. All changes made to the manuscript in response to the observations have been indicated and are marked in yellow. We hope the new version meets your expectations.
Comment: several tables have either "media" or "medium" when it should be "Mean" or "Median" (I can't tell from the data which is intended) - it needs clarification
Response: Thank you very much for that observation. The language error has been corrected; now the term "mean" is used correctly throughout the article. Other language errors have also been reviewed.
Comment: the breakdown of ages (<28, 28-34, 35-43, and >42) seems artificial. they're not even intervals in terms of years, and they don't yield even quartiles. choose one or the other.
Response: We understand the concern regarding the unequal width of the intervals. However, since there are no clinically standardized age brackets in the context of ART, the age categories (<28, 28–34, 35–42, and >42 years) were defined based on the actual quartiles of our sample’s age distribution. Although the year ranges are not uniform, this approach allowed us to divide the population into four subgroups of approximately equal size, ensuring balanced comparisons across groups. This can be seen in the results (Table 3), and it is also now better described in the methodology section. Quartiles are: Q1=28, Q2=35 and Q3=42.
Comment: the statement that the AI goal is ≥95% is misleading; clinically the goal is viralogic suppression of ≥95% and you have that in your population, so the entire premise of the paper is not really appropriate. you should really be celebrating the fact that Biktarvy is performing so well (as it does, everywhere that it is used, due to ease of dosing, low toxicity profile, and high resistance barrier).
Response: We agree that virologic suppression is the ultimate clinical goal, which is observed in our results. However, our analysis also focuses on behavior towards adherence and its association with clinical data. In that regard, the ≥95% adherence threshold aligns with national guidelines. Even with high rates of virologic suppression, behavioral adherence monitoring remains relevant to ensure therapeutic success. We have clarified this in the abstract, discussion, and conclusion.
Comment: Figure 3 - the legend simply restates the X-axes of each plot, rather than expanding and explaining. Not helpful in understanding.
Response: Thank you for this observation. The legend for Figure 3 has been restructured to better explain the results.
COMMENTS ON THE ATTACHED DOCUMENT:
Comments on introduction:
Comment: Adherence has always been stressed at all visits everywhere that I have worked, and is never reserved as a discussion only to be conducted when a patient is failing.
Response: Regarding your concerns about adherence monitoring as a universal clinical practice, unfortunately, the Mexican Antiretroviral Management Guide for People with HIV formally mandates adherence evaluation only in cases of virological failure, but, as you suggested, it does not mean that it is not usually measured. On the other hand, the ACTG Follow-Up Questionnaire is available in electronic format in Mexican institutions, but there are no official records of the measurement at the CSCs, nor any other public data. Indeed, as you have suggested, this does not mean that adherence is not measured, but that it is not officially recorded, which also highlights the importance of our results. This context has now been cleared out and specified in the introduction in a better way.
Comments on Table 5:
Comment: 51-100 copies/ml is unsuppressed. Maybe if you combine all individuals with any values above 50 copies to the ref of <50, you might see a significant relationship? That would be meaningful.
Response: This is a very good suggestion. We have now categorized viral load into only two categories, as you have suggested. Therefore, inferential statistical methods were carried out again.
Comment about 350-499 cell/μL group having significant results: not sure what this means, particularly when the 500 cell/μL group is not significant.
Response: It is a very good observation because it lacked a proper discussion. It is correct that there is a significant association observed in the 350–499 CD4 cells/μL group, but not in the ≥500 group, and we have now added a discussion to address an interpretation of this result: patients in the intermediate range might still perceive themselves at risk or under closer clinical observation, which could motivate stronger adherence behaviors. In contrast, those with ≥500 cells/μL may feel less urgency to strictly adhere, especially if asymptomatic.
Comment about the weekend suspension having significant results: this seems expected and unsurprising.
Response: Even though ART suspension on weekends could be unsurprisingly associated with non-adherence, it is important to note that this result is statistically significant, and, as it is stated later in the discussion, ART suspension on weekends was the only variable that was not included in the original equation developed by Raynolds (2007) to calculate AI. Therefore, our result contrasts with the equation construction and suggests that this may be a previously underestimated variable.
Comment on Figure 4: This is the expected result of treatment, it doesn´t really say anything novel.
Response: Regarding the Markov cohort, we acknowledge that the trends shown in Figure 4 may be expected under effective ART. However, the value of this graphic lies in describing real-world data and confirming expected immunological recovery patterns from the studied population using the developed Markov model. Additionally, the fact that the Markov cohort graphic shows expected results is good news, since our Markov model was constructed and fed with the aim of representing realistic immunological changes, which are indeed observed in the graphic. Furthermore, this graphic helps to discuss your next comment.
Comment on the discussion regarding PLwHIV not reaching 500 cell/μL: This isn´t surprising, particularly in a cohort that starts treatment at <200 cell/Μl. Arriving at > 350 cell/μL is clinically fine and not necessarily placing patients at risk of failing ill with OI, so no real reason to make a big deal of this. It´s more a result, I believe of starting late, than a marker of lower adherence.
Response: We agree that starting ART at a CD4 count <200 cells/μL strongly influences the likelihood of full immunological recovery. We also agree that > 350 cells/μL is already clinically favorable for PLwHIV. We have now clarified these points in the discussion. The Markov cohort (Figure 4) also supports the discussion of this point.
Round 2
Reviewer 2 Report
Comments and Suggestions for Authors
The manuscript is definitely improved, and I would recommend proceeding with publication in its current form.